# What Participation Creates in Experimental Design Practices. The Case of a Mobile Third Place Built in a Retirement Home

Marine Royer

Department of Science and Arts, University of Nîmes, 30000 Nîmes, France; marine.royer@unimes.fr

**Abstract:** This article explores the rise of a new generation of practices combining architecture, design, and art, trying to answer the transition issues faced by society. It develops original operating procedures, including public participation. In doing so, those so-called "specialised" professions expand their sphere of operation and incorporate more immaterial dimensions and resources. The main objective of the article is an attempt to clarify how participation is embodied in specific intervention methods, within those experimental practices. The article will take as a case study a participatory project taking place in a retirement home and aimed at building a mobile third place that brought together various professionals coming from those experimental practices. The study of the participatory project will outline three devices and methods supporting the participation work, as follows: the use of permanence, the use of the prototype and self-construction, and the conception of ephemeral production. The article suggests that based on their analysis, we can understand what architects and designers "manufacture" through the agency of participation. Or more accurately, what participation "manufactures" in those experimental practices. The main result of the article is that the participatory project is more concerned with the motives and aspirations of the design activity, its methods and processes, its context and socialisation than it is with what would be classically considered as the outcome or result (the work, the realisation, the production, the built).

**Keywords:** experimental practices; architecture; design; participatory construction project; retirement home; third place; France

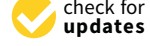



## 1. Introduction. Towards a New Generation of Practices Merging Architecture and Design?

There is a growing number of experimental experiences in design, architecture and art. They address transition issues raised by society, whether they are of an environmental, social, or sanitary nature. Their spread allows residents, professionals, and elected officials to take into their own hands those questions, locally and on a daily basis, through the shared design of projects related to public spaces, housing, facilities, and "infinite places" [1]. Those alternatives conditioned by major crisis disrupt design practices by tackling their innermost paradoxes: their industrial roots (in a context of a post-industrial crisis) and the destruction of ecosystems through mass-scale and continuous manufacturing of artifacts (in a world in the midst of an ecological crisis). Usually gathering as research programs, practical experiments, integrated laboratories, or multidisciplinary residencies, those groups are choosing as their testing ground the city and aim to reinvent people's daily lives.

Based on the definition by Herbert Simon [2], who considers design as the transformation of existing conditions into preferred ones, those experimental practices can be considered as a new sustainable generation of design. They have as a common feature to be working from a disseminated form of design, in its forms, scales, temporalities, and actors, as wanted by Victor Papanek [3]. The forms and formats of this new generation widen the field of practices traditionally regarded as belonging to architects, designers, or artists, and stand at the boundaries of those disciplines. Their hybrid classification relegates to the background the disciplinary dimension. In that respect, the experimental nature is less

about the practices than it is about the productions. It is about experimental architectures or experimental projects (performances, installations, objects, services) [4].

When it comes to design, those alternative practices are referred to under the terms of social, public interest, sustainable, territorial [5–7]. They uncover new territories through the prism of environmental prerogatives and the crisis of mass production. Even further removed from industrial artifacts than service design is, this field of design desires to model, and therefore takes as its material social interactions. Those actors consider that if design is responsible for the layout of the world, at every scale, it is its responsibility to respond to the challenges it faces. However, these practices bearing a reformist and even activist critic of the material conditions of our environments is not a novelty. Various currents of thought and production such as Italian radical design [8], or critical design [9] show that design can question the constructions of the world through what it produces and the way it does so.

In the architectural field, there is a form of collective practice of projects that seems to be emerging, led by a new generation of architects. Examples of this are the group exhibition "Re-Architecture" that took place in Paris in 2012 or the exhibition "Urbanités inattendues" in Toulouse in 2011. Those "inhabited" architects [10] form a heterogeneous network of practitioners gathered as a collective or a group who use atypical approaches to try to re-invent the traditional frameworks of project practice [11]. This collaborative work is not new, and often the architects have collaborated with one another and with other disciplines. It would be incongruous to consider that the architect acts alone. Some groups have made history by the continuity of their work under a common name, removing at least in appearance individualities. The era of the 60s and the 70s, for instance, has been especially prolific: Team X, Super Studio, Archizoom, Archigram, Coop Himmelb(l)au, AUA, Team Zoo, Ant Farm. As for the political significance of practices in architecture, it is not recent either, and can be traced back in currents such as experimental or radical architecture [12].

If those practices have an affiliation with certain design and architecture currents, it is their ways of proceeding that are unprecedented. They have developed original procedures such as: the involvement of users [13], co-creation practices [14], environmental influences whether they are urban, social, or natural [15], the processual character of production, the contextualised and located quality of the practices, user involvement [16]. In doing so, those so-called "specialised" professions expand their sphere of operation and incorporate dimensions and resources less used previously, or even neglected. They "de-specialise" their work matters and materials.

Furthermore, this raises a series of challenges for architecture and design. On one hand, those practitioners using original procedures struggle to find their legitimacy in their own disciplinary field. For example, in France if some of those practitioners call themselves "architects", and are graduates from a school of architecture, most of them didn't follow through with the license giving them authorisation to exercise architecture under its own name (HMONP), and therefore are not registered with the French Ordre des Architectes. In theory, they can't declare themselves as architects. This conflict of use of one's title is far from being anecdotal and encourages us to consider those practices as offering new ways to consider the trade and the training [17]. On the other hand, the methods that are implemented in the making of projects are still uncertain and exploratory. It should be noted that participation, just to take this example, is not a novelty in design practices, including in architecture. Ever since the 1970s, the definition of architecture is called into question with the introduction of social science in the teaching of architecture and with the first experiments of resident participation in development projects [18]. What is new is the forms taken by this participation. However, by becoming apparent, these "new" factors of production disrupt the competences. They are often less "objectifable", less concrete and material, and imply other knowledge and expertise. They are at least for now, less defined and well-characterised, as they are de facto less professionally established. They are more difficult to work on because of this dual character, more "immaterial" [19] and

less established. Among them, the article proposes to focus on the participation of people concerned. As a matter of fact, studying those projects that are situated and built for a purpose, questions those ways to act through which the public is invited to participate in order to bring out new desires of a common space. It raises the following questions: how to drive participation? How to objectify a co-creation practice? How to assess the relevance of such methods?

The main objective of the article is an attempt to clarify how participation is embodied in specific intervention methods, within those experimental practices. The notion of "participation" encompasses multiple realities: consultation, dialogue, resident participation, delegation, co-production, participatory democracy. Participation, as an umbrella term, has entered the vocabulary and practices of a significant number of actors, often conflicting, competing for its meaning, purpose, and value [20,21]. It is therefore important to go over in detail the participatory processes implemented in these practices to understand their meaning and function. Is it about getting the residents ready for the idea of the project? To foster their adhesion to it? Or is it about something else?

For this purpose, the article will take as a case study a participatory project taking place in a retirement home and aimed at building a mobile third place that brought together various professionals coming from those experimental practices (designers, architects, and artists). The notion of "third place" was introduced by Ray Oldenburg in order to comment the birth of new, intermediate, and "in-between" (home and work) places. Places where a community life crystallizes allowing broader exchanges on a more local level [22].

The study of the participatory third place project will allow us to outline three devices and methods supporting the participation work, as follows: the use of permanence, the use of the prototype and self-construction, and the conception of ephemeral production. The article suggests that based on their analysis, we can understand what architects and designers "manufacture" through the agency of participation. Or more accurately, what participation "manufactures" in those experimental practices in architecture or design. The main result of the article is that the participatory project is more concerned with the motives and aspirations of the design activity, its methods and processes, its context and socialisation than it is with what would be classically considered as the outcome or result (the work, the realisation, the production, the built).

## 2. Research Methods. Action Research through Experimentation

Research in design or in architecture is being carried out in pluralistic and heterogeneous approaches. The research the article is based on favours an approach of hybrid research that can be defined as Action Research (A-R) through experimentation. This is a scientific research approach and methodology which is aimed at conducting in parallel and in an intricate manner the acquisition of scientific knowledge with concrete and transformative actions on the ground (via experimentations). This dual objective belongs to a larger tradition in social science that can be attributed to the work of Kurt Lewin [23] and can be defined as follows: "Action research is a fundamental research approach in human sciences that arises from the encounter between a desire for change and a search intent" [24] (p. 87). It is not research in the service of action, but action (or intervention) producing elements of research [25]. Nevertheless, it can still be considered today as an "unclear concept" [26], as there are several nuances in research positions, ranging from simply taking into account the challenges faced by people on the ground, all the way to the participation of the anthropologist in advocating for a cause [27] or the construction of a policy [28]. This research method does however share a common feature: no longer does it just describe a pre-existing reality, it accompanies inevitable changes.

The action research through experimentation described in this article originated in 2020, when a reflection was initiated to create a third place in the future Centre for Gerontology of Nîmes (CGN) in the South of France. The third place would be dedicated to "social connection, citizen engagement and the ageing-well plan on it territory" (extract from the general diagnostic document). Such a space of shared and open activities has

the following objectives: to promote the inclusion and participation of the elderly and their families; to open the gerontology center to its environment by making it a social and cultural actor in the neighborhood; to develop a model of a third place adapted to the medical social challenges.

The third place embodies the will of the CGN to rethink its practices around the well-being of its residents and employees, as well as the position of retirement homes towards inhabitants and the dynamics shaping its territory. Consistent with the evolution of public policy in France in favour of mechanisms fostering participation, inclusion, and citizenship of the elderly (reports Libault on "Advanced age and autonomy" and El Khomri "Attractiveness of late-life jobs" (2019) and Guedj report on isolation of the elderly (2020)). Following unsuccessful negotiations to acquire land for the construction of the building, the project of a third place became inscribed in a double temporality. On one hand, one of the constructions of a "physical" and permanent third place in a dedicated space within the future CGN (timeframe 2023–2024). On the other hand, since October 2020, a feasibility phase of the third place "outside the walls" that should allow to bring together a community around the project, to come up with events programming and put in place activities in relation to local actors.

To understand and analyse the context, the action research through experimentation has favoured participant observation and qualitative interviews. This inductive approach [29] can provide two types of research findings: on one hand, the participation of researchers committed to the development of a real alternative [30] through the creation of a mobile third place; on the other hand, through the study of data collected on the ground, the acquisition of scientific knowledge regarding the methods and tools of participation in the experimental design practices.

The choice to carry out a so-called "simple" case study in this article, which is a study analyzing only one project, is justified for various reasons, in particular by the fact that it is a "representative" or a "typical" case, which is especially revealing of situations common to experimental practices [31].

The three methods of participatory interventions presented in part 3 stem from the analysis of different projects within exploratory design practices. They are also interpreted from the perspective of previous studies (cited throughout the article, including [4,10,11,13,16,17]), and the Ph.D. thesis of a member of the collective of architects presents on the participatory construction project [12].

## 3. Results. Three Methods of Participatory Interventions in Experimental Design Practices

The founding moment in the design of the mobile third place was the participatory construction project that took place in March 2021 and that aimed at developing from a Peugeot J9 vehicle a mobile and intergenerational space to build a bridge between the retirement homes and the projects (Figure 1). During two weeks, the participatory project gathered "builders-architects" (the collective Etc.), an artist (Bonnefrite), a sociologist, a researcher-designer, students in their first year of university, elderly people, their caregivers, retirement home professionals and residents of Nîmes. The designers suggested involving the community of interest of the future third place as early as the construction phase of the work to foster their involvement, in order to create a self-managed situation. The devices and methods implemented to engage and integrate the people involved in the making of the mobile third place are observed in the remainder of the article and thus characterize the participation.

### 3.1. Participation to Supply the Processual Character of Production: The Permanence

Residing on the location of design is one of the common methods of intervention in experimental practices. A fairly recent concept, architectural permanency or design residency allows designers to settle in spaces over a long period of time, to understand the complexity of the territory on which they will intervene. It is also an opportunity to involve the people on said territory in the construction of the work. For architects, residency was

democratised in the 2000s thanks to the Construire workshop [32]. Borrowed by architects to the world of performing arts, where artist residencies are frequent, this principle is also affiliated with the tradition of participation in architecture, notably carried by Lucien and Simone Kroll [10].

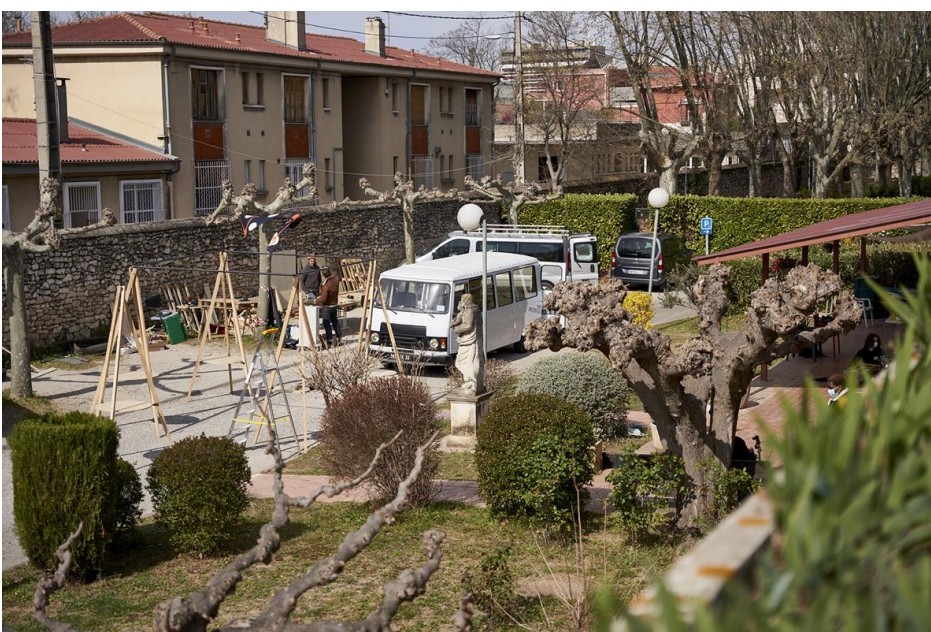

**Figure 1.** Photography of the yard of the retirement home and the J9 vehicle during the participatory project, Nîmes, 2021© Thomas Heydon.

The Atelier Kroll has a very significant place in the history of architecture for its participatory approach since the 1960s. Their most famous work is the Mémé (Maison médicale des étudiants en médecine), medical faculty housing, designed and built between 1970 and 1972. It was designed based on discussions with the medical students and some months-long effort on the blueprints that were formed, modified, completed, little by little, by the desires of some, the ideas of others, the needs of some, etc. (Figure 2).

It is this labour through strata, incrementation, that shaped the building. Patrick Bouchain talks about this project in these terms: "when I saw this work, I understood that an architecture of diversity and disagreement could exist, a truly democratic architecture." [10]. One of the pillars of the Atelier Kroll's work is to participate not "with" or "for" the residents, but "as one" of the residents [33]. The architect Lucien Kroll introduces himself above all as a citizen, a resident, and works from his own experience of inhabiting.

In the approach of the team to the mobile third place, the participatory construction site is transformed, in the same perspective, into a meeting point, where the installations under construction are lived, where the visitor becomes an actor, where the location becomes a laboratory of experimentation, but also of exchange, life and work. The construction site was installed in the yard of the retirement home, where residents could stop, observe or take part in the activities. On one hand, it allows for the engagement of the audience of the elderly, that is rather captive but also constrained by fragmented days (tiredness, fluctuating motivation, interruptions due to caregiving, etc.). On the other hand, the construction site allows the retirement home to open up to the outside by inviting other people (students, neighbours, associations, elected officials). For the practitioners (architects, artists, designer-researchers, design students), it is no longer a question of involving, but more one of being involved themselves in the extensive process of designing: to live within the context to understand its specificities and make the design project evolve accordingly.

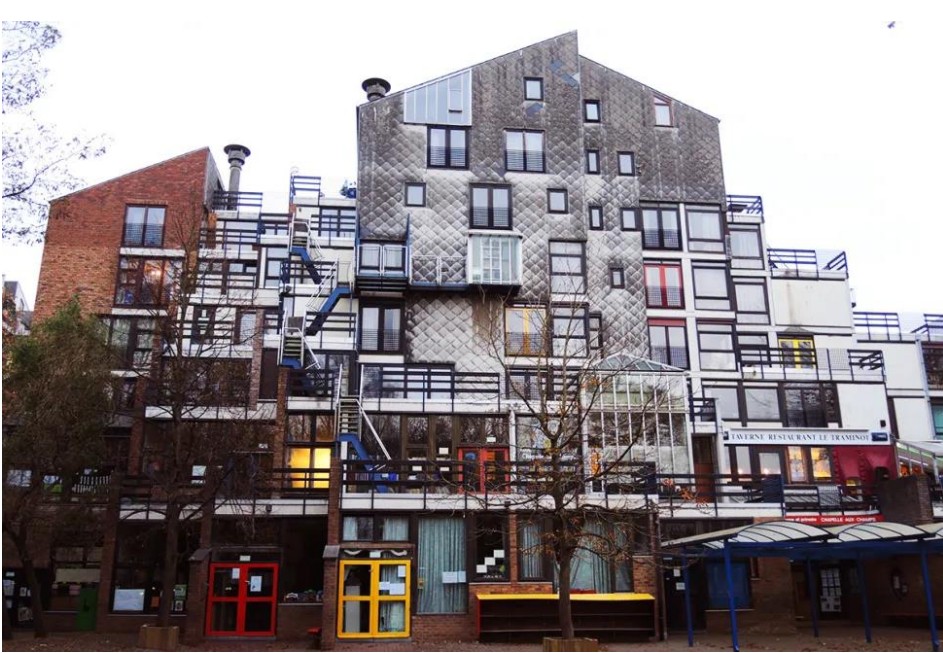

**Figure 2.** Photography of La mémé—medical faculty housing of Simone and Lucien Kroll, Louvain-la-Neuve, Belgium© Estelle Sauvaitre.

### 3.2. Participation in Order to Collectively Shape the Work: The Prototype, the Self-Construction

Those alternative practices carry a radical push to "do" differently. The second method of participatory intervention is to bring to life this project of experimentation on full-scale, whether it is about building a prototype for a service or creating bigger spaces. As opposed to conventional design processes, where the designer draws and the builder constructs, those professionals bring together design and construction. The idea is to construct while continuing to design on site, by rediscovering a form of building intelligence in the materials and expertise available. Those practitioners are less committed to the construction techniques and the methods used than they are to give meaning to a common approach that allows each and every person to take over and use techniques.

The participatory construction site is conceived as a combination of building workshops for the structure (poles, external venetian, etc.) and the furniture (Figure 3), but also workshops for sewing (Figure 4), painting, graphic designing, in order to decompartmentalise skills and design the work collectively. The equipment proposed by the architects is simple and easy to use: saws, screw drivers, non-professional sewing machines, paint brushes, etc. Those experimental practices have as a core idea that the practitioner must master the tools and being able to pass on how to use them.

In the way of making, as well as the symbolic aspect, those practices bring to mind do-it-yourself handiwork. The formal proposal of the mobile third place project summons the imagery of a shack, a scout camp, suggesting that another way to build the city is possible, using self-construction. This approach seems above all of a political nature, to circumvent the traditional logic of production.

Alongside those workshops, the artist Bonnefrite organises some "bals-peinture", paint dances (Figure 5). The idea is to invite the residents wishing to do so to contribute to the graphic design of the denim canvases that will be used in all the textile elements of the project, and by doing so during a convivial gathering. In order to create paint tools that are adapted to the residents' motor skills, we create them together, using simple and inexpensive means, by recycling former medical devices that had been discarded. In that instance as well, auto-construction and handiwork generate a collective imaginary referencing the willingness to hijack the traditional logic of production of artworks. The practice of self-construction lies in the fact of composing with the existing, by arranging

with fresh eyes the shapes, know-how, available materials to foster the appropriation of works and their co-productions.

### 3.3. Participation to Build Durable Events: Ephemeral Artifacts

The experimental nature of the participatory construction site is clearly assumed and even claimed by the observed practitioners. Their activities show more of a discovery than an affirmation, more of an experimentation than a programming. This induces in projects a specific relation to time, more ephemeral than in conventional projects. They even transform the status of the work, by not producing architectural or design projects in the strict sense, but "events that are built for the time of action to reveal potential usages and to raise awareness in residents on the definition of their living environment [34] (p. 56). The short temporality seems to allow trial and error (preferred method of design) all while giving a festive character to projects (Figure 6).

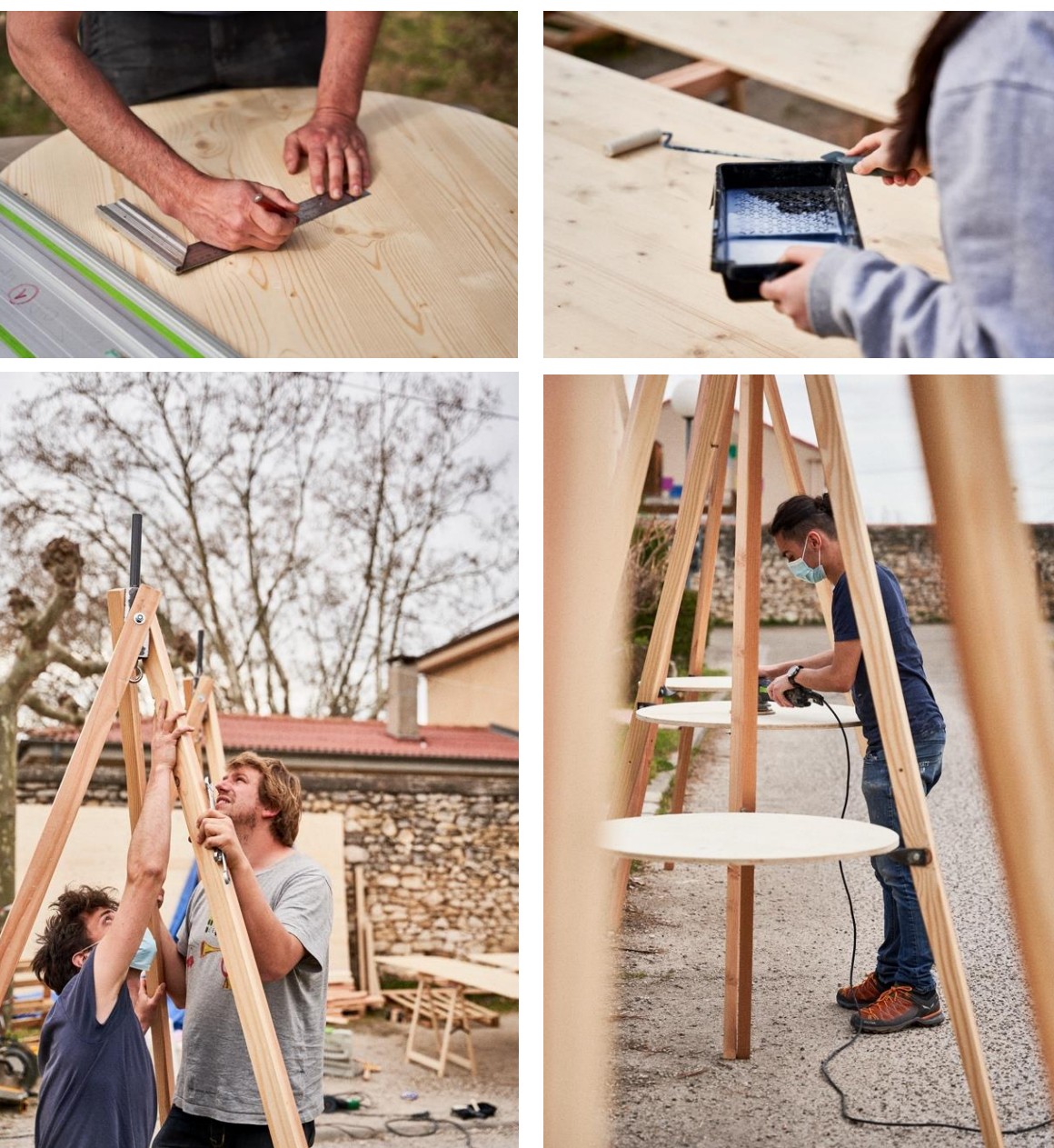

**Figure 3.** Photographs of the first week of construction in the retirement home, construction of totems in wooden cleats with the students Nîmes, 2021© Thomas Heydon.

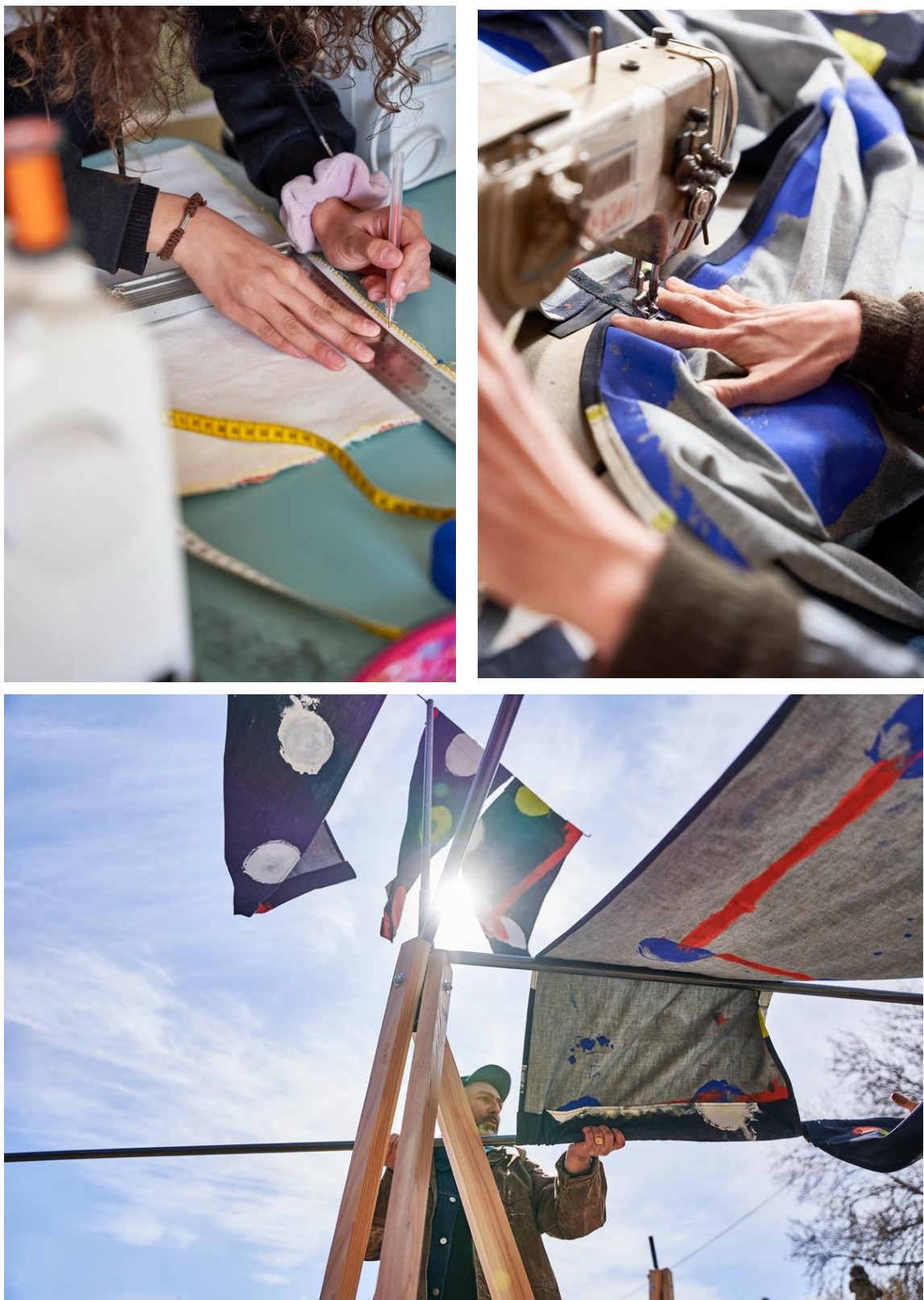

**Figure 4.** Photographs of a sewing workshop of the elements of fabric of the project with volunteers and residents of the retirement home, Nîmes, 2021© Thomas Heydon.

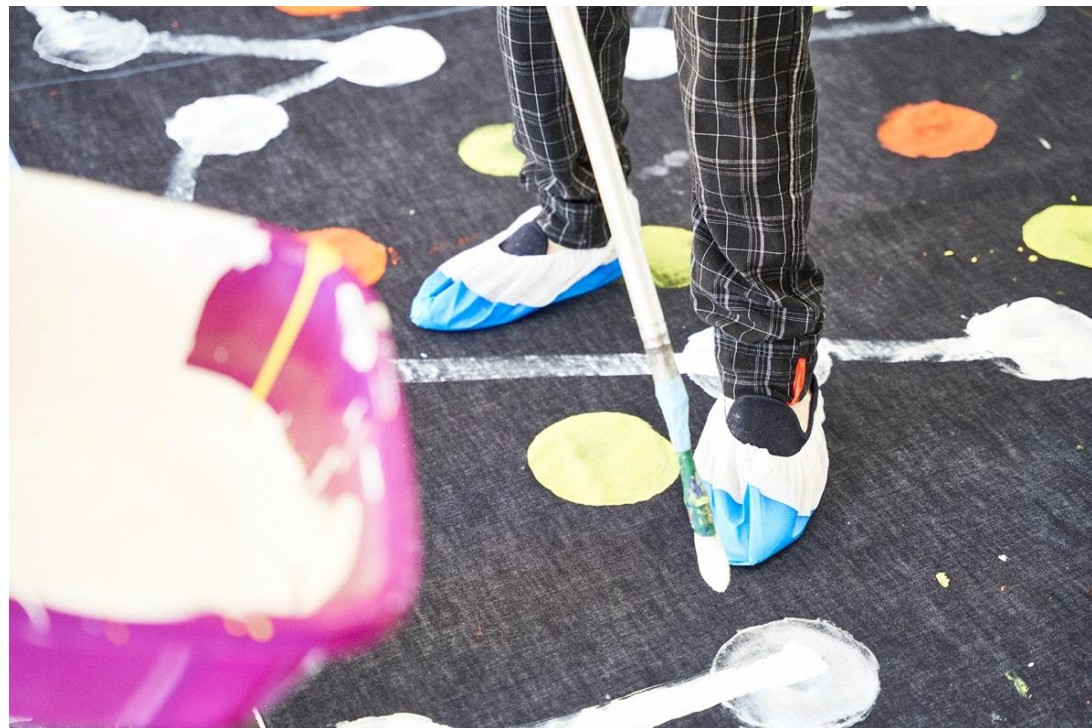

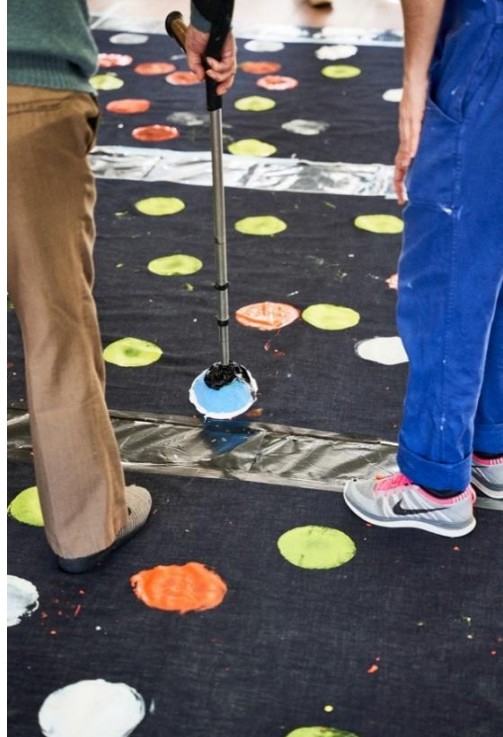
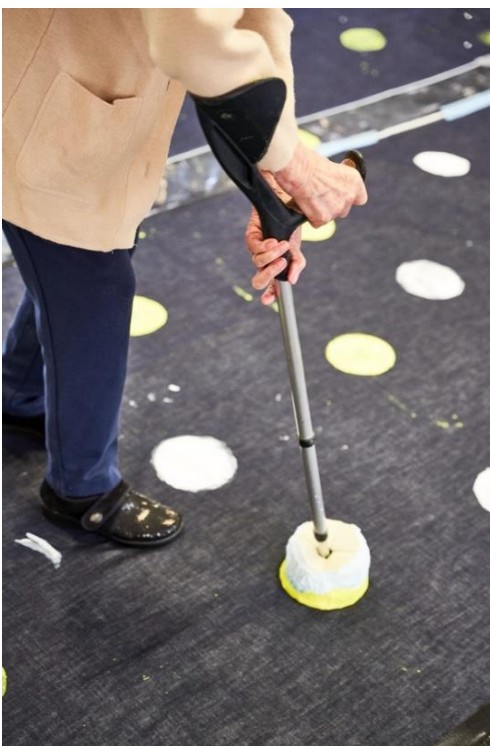

**Figure 5.** Photographs of the first "bal-peinture" at the retirement home, Nîmes, 2021© Thomas Heydon.

The ephemeral durability of the design projects is an essential factor in experimental practices. Anne-Marie Lecoq, when talking about ephemeral architecture, points out two elements to the analysis of the effects of this factor. On one hand, it lies in "the absence of imperatives of robustness, hence the choice of perishable materials and specific construction techniques" [35] (p. 437). By using materials and techniques of simple construction, this first aspect allows practitioners to share their expertise with the people on the construction sites, from a participatory perspective. The structures of the mobile third place are, for

example, made of wooden cleats. On the other hand, and paradoxically so, Lecoq claims that ephemeral architecture is the "opposition to the temporary", because it is not "a makeshift solution while waiting for something else, it is a structure that exists for itself, until something happens" [35] (p. 437).

The mobile third place has been operational since March 2021 (from the end of the participatory construction project). It is now used in the organisation of local events bringing together retirement home residents and citizens: artistic workshops during an illustration festival, meetings and debates in a municipal park, moments of awareness on the old age in a square, etc. Its uses depend on the themes chosen and the people involved. In this context, the structures made of wooden cleats will be used until the exhaustion of the material (until they break, damage, wear out). The structures can be replaced, repaired, then thrown away when they are too damaged.

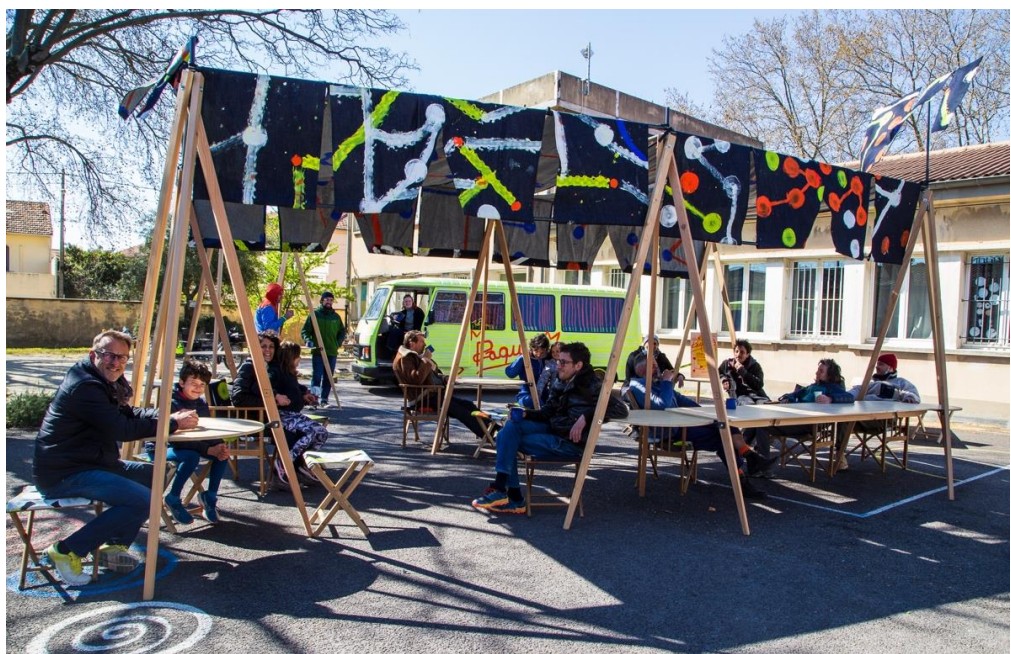

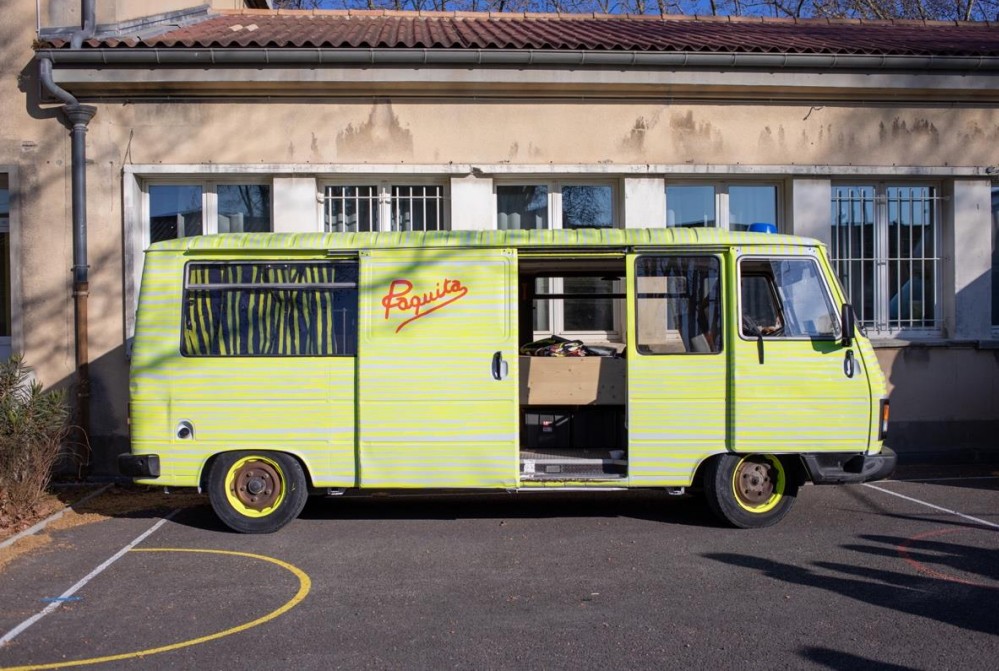

**Figure 6.** *Cont.*

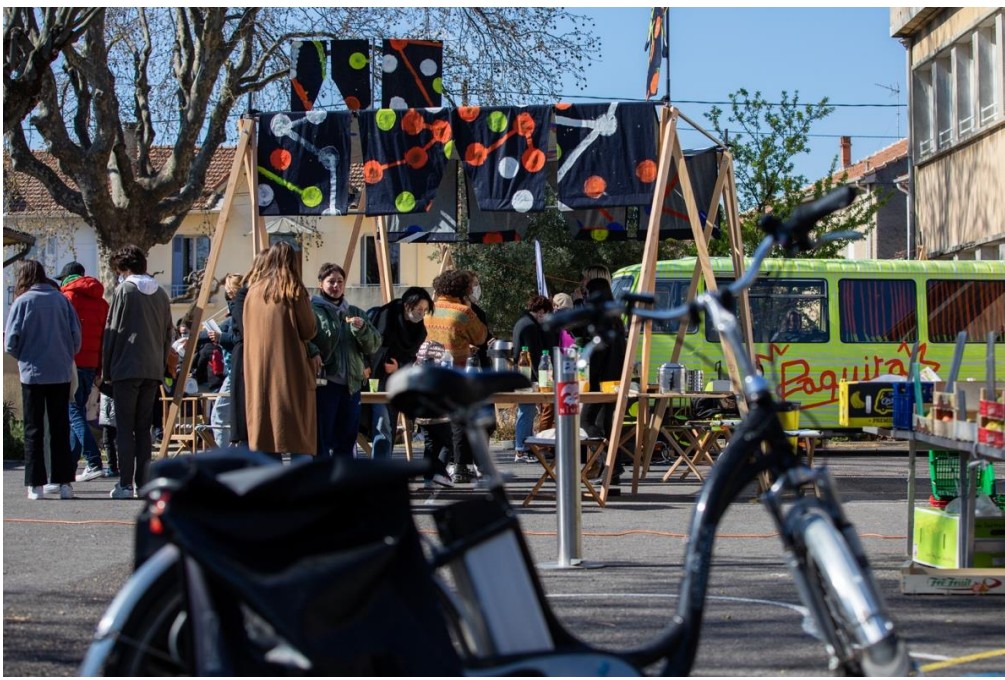

**Figure 6.** Opening of "Paquita" the mobile third place of the Centre for Gerontology of Nîmes, Market of Beausoleil, Nîmes, 2021© Thomas Heydon.

During the participatory project, the practitioners defend the idea that the ephemeral can be a token of durability or longevity. Without duplicity, this affirmation advocates for reversible works, "mutable" [36], that, given the complexity of contexts, appear to them as more sustainable than the quest for infinite duration that is pinned to the architectural discipline. They engage in their activities with the awareness of not knowing, or at least, not knowing enough about the challenges they face. Those practitioners are more preoccupied by what is taking place and the processes involved.

## 4. Discussion. From Participation to Involvement: Building the Immaterial

Through the analysis of those three methods, the work of public participation can be more accurately described as a work of involvement. This distinction between participation and involvement finds a conceptual echo with the opposition that Éloi Laurent offers between the terms of "cooperation" and "collaboration" [37]. Two particularly striking dimensions separate both notions and make it possible to understand the contributions of this methodological and conceptual approach:

Firstly, collaboration is carried out solely through work, whereas cooperation calls on all the human capacities and purposes. As part of the participation of elderly or vulnerable people, the idea was to offer activities in "open" formats, such as "bals-peinture". Indeed, the modes of cooperation of the elderly people were not defined in advance. Some were actively involved in the painting activity. Others stood in the background, listening to the music or chatting in the middle of this merry mess. Depending on the people's abilities (motor, psychic, communication), the ways of cooperating were different, but equal.

Secondly, collaboration is an association with a specified purpose, whereas cooperation is a free process of mutual discoveries. The participatory construction project has made it possible to initiate cooperation between people who are now part of Paquita community, the mobile third place. The elderly and students who have been integrated since the construction are now human resources for the truck's activities. The programming of the third place is not fixed, it changes according to the desires of the people. The structures designed by the architects allow different uses: sitting or standing, alone or in a group, working, eating, projecting a film, etc.

For the experimental design practices observed, the aim is to design concrete objects in order to create situations of collective involvement that outlast the construction project. Therefore, it can be observed that the concrete manufacturing (of a space, a service, an installation, etc.) becomes the pretext of a more immaterial manufacturing, in order to create a self-managed situation, a community that transcends the design project of architects and designers. It would appear that with involvement in architecture or in design lies one of the profound motives that explain the solicitation and mobilisation of those new resources. It seems that a construction "in itself", a build "in itself" are no longer sufficient to legitimise a practice, for the future users of the structure as well as for its developers. Those two observations are startling and could usefully be investigated.

On one hand, this construct needs to make sense, and to do so in situation, in context, especially for people. It also needs to do so for environments, for places, and ultimately, for lives. It is often assumed that consultation is firstly for the benefit of people, who ask to be informed, but also to understand, and sometimes take hold of projects that transform their environment. It is less often considered that the approach contributes to the collective reflection. The expression of "user's experiences" enabled the residents to have a say in the approaches to projects linked to the city [38]. But paradoxically, this formula also reduced this seat at the table as one of experts of the daily life: "tell us about your commuting, your habits and your expectations in terms of usages". However, the expectation of the residents is much more ambitious: they want to debate and act upon the social and political issues underlying urban projects [39]. The hypothesis of the studied practitioners is that the understanding and the debating of these issues can only be addressed by the concrete and collective manufacturing of these projects. If the "doing" in the sense made by Michel Lallement [40], is questioned, if the ways of doing are experimented with differently, if the arts of doing are strongly re-engaged, then meaning is restored to the idea of resident involvement. However, a first limitation must be noted here. This methodological and conceptual approach favoring intergenerational involvement and cooperation rather than a construct "in itself" is not reassuring for sponsors and funders. It is difficult to assert the experimental dimension of these projects according to which it is possible to make a mistake, the "right to error".

Furthermore, it appears that this construct needs to make sense to practitioners themselves. The integration of new resources, including the involvement of people, transforms significantly the practice of the trades and the scope of professional expertise. The architect and the designer are moving towards records of experiences that historically don't belong to their domain [19]. Another limitation is identified here. This concerns the capacity of these practitioners to "institute" their methodological and conceptual approach in order to stabilize it and allow other professionals to use it. Indeed, "new" factors of production have appeared. They are unusual and trouble the usual skills. They are often less "objectivable", less material or concrete, and therefore require other know-how and expertise. They are, in any case, for the moment less defined and less well characterized because, in fact, less "established" professionally. The approach described in the article is difficult to grasp because of this double character, more "immaterial" [19] and less established.

## 5. Conclusions

This article demonstrated that experimental design practices offer a choice of situations that define themselves at the intersection of architecture, design, and art. They take shape in temporary and reversible constructions, art installations, participatory processes, third places that self-construct in a logic of appropriation. They appeal to original operating modes, such as public participation. Three methods of intervention supporting the participatory dynamics of those practices that have been presented, coming from the study of participatory construction site gathering some of these practitioners. Firstly, the article presented the act of maintaining a permanence on the construction site in order to involve the future users of a project all through the different phases of the design process. The permanence also allows the practitioners to switch position and to modify their design

intentions according to the advice, resources, and expertise available on the ground. Secondly, the article unveiled how participation of people requires the appropriation and usage of simple construction techniques in a self-building dynamic. The idea being to generate activities of co-production of the work with those present. Finally, the politics of time of those experimental practices have been observed. It relates to the present time, to an emerging becoming, rather than to a desired or possible future. It is then a question of building "events" to reveal the potential usages of the places and show that another way of constructing living environments is possible.

For the mobile third place, the aim is to design concrete objects in order to create situations of collective involvement that outlast the construction project and become the vector of a more immaterial manufacturing. This partial dematerialisation of the work subject can be perceived as an expansion of their experimental ground and field of competencies, a way of manufacturing design projects "with a renewed know-how [ . . . ] on new grounds, conducive to innovation" [41]. Furthermore, the creation of forms becomes, at the time of manufacture, a fundamental issue in debates with users, touching the very core of the trade of the architect or designer. Anne Querrien defines the stake of the "architectural statement" as follows: "the possibility for a building to escape mastery of its designer, and to simply become a living place" of which the creative process would be taken care of by its users [42]. The aesthetics contribution of those experimental design practices to the fields of architecture and design as part of co-production approaches remains to be studied, as it appears essential.

**Funding:** This research was funded by La Conférence des Financeurs de la Prévention de ja Perte D'autonomie [CFPPA], call for projects July 2020.

**Institutional Review Board Statement:** The study was conducted according to the guidelines of the Declaration of Helsinki and approved by the Institutional Re-view Board of La Conférence des Financeurs de la Prévention de la Perte D'autonomie [CFPPA].

**Informed Consent Statement:** Informed consent was obtained from all subjects involved in the study. Written informed consent has been obtained from EHPAD residents to publish this paper.

**Data Availability Statement:** Not applicable.

**Acknowledgments:** I wish to acknowledge the support provided by all the staff at two retirements homes in Nîmes for their invaluable help, before and during the participatory construction project. It involved hard work, with moments charged with great emotional intensity and some interesting findings that would not have been possible without the help of all the staff. I would also like to thank the team: the collective Etc., Bonnefrite, François Huguet, the University students, the residents and their caregivers. Each of them participated, at their level, in the construction and success of the project.

**Conflicts of Interest:** The author declares no conflict of interest. The funders had no role in the design of the study; in the collection, analyses, or interpretation of data; in the writing of the manuscript, or in the decision to publish the results.

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
