# Peer review of "What Participation Creates in Experimental Design Practices. The Case of a Mobile Third Place Built in a Retirement Home"

_2673-8945, doi:10.3390/architecture2010004_

Round 1
Reviewer 1 Report
The paper discusses a recently emerged phenomenon of combining art, architecture and design in a single social event. A resident design case study is reported based on a retirement house event. The paper is well-written and interesting to read. Moreover, it is very touching and the underlying concept is well-grounded through a careful literature review. I especially appreciate the artistic and philosophical basis of the study. Please, find some comments below.
Does globalized capitalistic system really exist? I believe there is no unipolar world anymore, and the globalization processes stopped and even reversed more than a decade ago.
How does the incorporated art & architecture work in 2nd and 3rd world countries?
Are the reported practices able to exist in socialistic states like China or Cuba? Are the proposed approaches applicable to local societies? If the proposed technique is generally applicable, I can recommend removing political arguments from the introduction.
Please, emphasize the crucial moments of socialization between the elders and the young people during your study. Does this interaction add some value to the overall results? Which interesting discoveries surfaced during the co-working? I also recommend adding more graphical examples of the created objects to the paper.
Can the described "concept of construction while continuing to design" be a form of research-based design practice well-known from technical sciences? When the properties of the object under design are unknown, and the developer invents and researches during the design. If yes, please, mention it in the literature review.
Do the described practices actually close to building anything useful, or were they mainly "ephemeral"? I like this duality between temporal and permanent as a concept, but such events consume real resources.
How were the developed structures utilized after the event? I believe each project should possess a lifecycle, including utilization & recycling. Ecological aspects are of great importance today. Moreover, the destruction of the constructed site as a part of everything that exists, can complete the philosophical concept of "ephemeral art & design".
Nevertheless, I highly like this elegant study and find the underlying idea of great interest. Therefore, I can recommend the paper for publication after only minor edits.
Reviewer 2 Report
The general approach of a design approach, which aims to modify given situations and relies on participatory strategies, is explained. However, it is not explained where the three methods as focused on come from. Instead, they are set and then illustrated by a single concrete project. The conclusion then seems to confirm the beginning of the essay. Apart from the fact that it is questionable whether, starting from a single project, generalizations can be made at all without at least making comparisons to other projects: It should at least be addressed whether the project succeeded in improving the situation of the elderly through the means employed. Did it succeed in encouraging participation, integrating the elderly or opening up the old people's home to its surroundings? Where were there obstacles or difficulties? At what points did the project go beyond the usual strategies of other participatory projects? Above all, the term "(mobile) third place" would have to be defined, taking into account the research literature.
Round 2
Reviewer 2 Report
I think the essay is now okay.